# Introducing Robust Statistics in the Uncertainty Quantification of Nuclear Safeguards Measurements

**DOI:** 10.3390/e24081160

**Published:** 2022-08-19

**Authors:** Andrea Cerasa

**Affiliations:** European Commission, Joint Research Centre, Via E. Fermi 2479, 21027 Ispra, VA, Italy; andrea.cerasa@ec.europa.eu; Tel.: +39-0332-789792

**Keywords:** robust statistics, nuclear safeguards, one-way random effect model, ANOVA, Q-estimator

## Abstract

The monitoring of nuclear safeguards measurements consists of verifying the coherence between the operator declarations and the corresponding inspector measurements on the same nuclear items. Significant deviations may be present in the data, as consequence of problems with the operator and/or inspector measurement systems. However, they could also be the result of data falsification. In both cases, quantitative analysis and statistical outcomes may be negatively affected by their presence unless robust statistical methods are used. This article aims to investigate the benefits deriving from the introduction of robust procedures in the nuclear safeguards context. In particular, we will introduce a robust estimator for the estimation of the uncertainty components of the measurement error model. The analysis will prove the capacity of robust procedures to limit the bias in simulated and empirical contexts to provide more sounding statistical outcomes. For these reasons, the introduction of robust procedures may represent a step forward in the still ongoing development of reliable uncertainty quantification methods for error variance estimation.

## 1. Introduction

Nuclear safeguards involve measures designed to deter and detect the diversion of nuclear material from the fuel cycle for illicit purposes [1]. The monitoring for possible data falsification by operators that could mask diversion is based on the statistical analysis of paired (operator, inspector) verification measurements. Significant deviations between the operator declarations and the corresponding inspector measurements may be the consequence of problems with their measurement systems. However, they could also be the result of the operator falsifying the data. In this context, the fundamental elements for a trustworthy analysis are *(i)* a proper measurement error model and *(ii)* an efficient and unbiased uncertainty quantification (UQ), i.e., the estimation of its uncertainty components.

Concerning this second point, two approaches are available: the bottom-up and the empirical top-down approach [2]. The former consists on quantifying and then propagating the uncertainty in all key steps of the assay, in order to derive the resulting uncertainty of the measured values. The latter, instead, relies fundamentally on modeling measurement data as a one-way random effect model with two variance components, one accounting for the variation within and the other for that between groups. A group is typically defined as an inspection period, within which the measurements are assumed to have the same random error. Between inspection periods, instead, the systematic error captures the changes that may occur in the metrological conditions, such as instrument calibration, change of inspectors, change of background radiation, and so on [3]. The two variance components with data from past inspections are estimated through the classical analysis of variance ([ANOVA], [4]), or using the Grubbs estimator [5], depending on the scope of the analysis.

In empirical application, the outcomes of the bottom-up and the empirical top-down approach could significantly diverge. Specifically, bottom-up estimations are often larger than their top-down counterparts [6]. The need to increase consistency between these two results has been stressed more than once (see, for example, [7,8]) and encouraged the study of potential improvements for both methods [9,10]. The proposed developments of top-down methods never took into consideration the possible presence of anomalous data in the estimation sample even though, as previously mentioned, significant deviations between operator and inspector measurements may occur due to multiple reasons.

The presence of outliers in the data may in fact seriously affect the reliability of the estimations. In random effect models, even one single anomalous observation may greatly influence the properties of the statistics [11]. The bias introduced by a single anomalous value will also affect the result of the subsequent analyses based on the between and within group standard deviations, as the setting of alarm thresholds in measurements (see, for example, [12]) or the material balance evaluation [1]. Consider, for example, the simulated case study in [12] (see Figure 1a). The 30 observations, equally split in 3 groups, are simulated using a random effect model with between and within group standard deviations both equal to 0.010. They mimic 30 possible relative distances of the measurements provided by the operator and the inspector in three different inspection periods. Classical analysis-of-variance expressions provide respective estimates of 0.009 and 0.007. Suppose now to change only one single value and convert it in an outliers, as we do with the observation highlighted in red in Figure 1b. This small change yields a remarkable decrease of the sample average of the 10 observations of group 1, and the classical ANOVA estimates for the between and within group standard deviations become respectively 0.003 and 0.012. It is important to stress that the 30 simulated observations cannot be considered as a time series, even if their plots may suggest it. This is because they are mimicking a use case in the nuclear safeguards context, where the time lag between two single inspections and even between two groups of inspection is not fixed and constant. Therefore, methods coming from time series literature cannot be employed in this context.

The aim of this article is to propose an alternative approach for the UQ problem through the introduction of robust estimation of the between and within group variance estimators. The robust analysis is expected to limit the influence of outliers in past data on the outcomes and to improve the quality and the reliability of the statistical outcomes. For this purpose, we will compare in a simulation exercise the accuracy of the classical estimator with respect to one of the many methods available for estimating the variance components of a random effect model. In particular, we focus on the robust estimators proposed by [13] and based on the Q-estimator for one-sample scale introduced by [14], whose main strengths are:*Suitable breakdown point:* they have the capacity to handle a consistent proportion of outliers in the data before returning an incorrect result;*High efficiency:* they provide estimates with desirable properties even when there are no outliers in the data;*Closed expression:* they do not depend on optimization algorithms and are rather easy to calculate.

Thanks to these properties, the proposed estimators are extremely flexible, and may represent a step forward in the still ongoing development of reliable UQ methods for error variance estimation [9].

This paper proceeds as follows. In the next section, the measurement error model and the classical estimation of its variance components are introduced. Then, in Section 3, the alternative estimation method based on robust statistics is described. In Section 4, the results of a simulation experiment that analyzes the behavior of the two estimators subject to different contamination scenarios are presented. An empirical application of UQ in nuclear safeguards measurements through classical and robust estimates is described in Section 6. Finally, Section 6 concludes.

## 2. Measurement Error Model and the Classical Estimation of Its Variance Components

Given a set of data from past inspections and assuming the same number of measurements in each inspection (i.e., balanced design), a typical model for representing the inspector (*I*) and operator (*O*) measurements of the item *k* during inspection *j* is the following:$$I_{jk}=\mu_{jk}+s_{Ij}+r_{Ijk}\;\;\;\;O_{jk}=\mu_{jk}+s_{Oj}+r_{Ojk}\;\;\;\;j=1,\cdots ,g\;\;\;k=1,\cdots ,n$$
where $\mu_{jk}$ is the (unknown) true value of the measurand, $s_{*j}∼N\left(0,\sigma_{s*}^{2}\right)$ are the short-term systematic errors and $r_{*jk}∼N\left(0,\sigma_{r*}^{2}\right)$ are the random errors. Therefore, the difference between operator and inspector measurements is given by:(1)$$O_{jk}-I_{jk}=\left(s_{Oj}-s_{Ij}\right)+\left(r_{Ojk}-r_{Ijk}\right)=s_{j}+r_{jk}$$
which is a random effect model with variance components equal to $\sigma_{s}^{2}=\sigma_{sO}^{2}+\sigma_{sI}^{2}$ and $\sigma_{r}^{2}=\sigma_{rO}^{2}+\sigma_{rI}^{2}$.

If the error is expected to scale with the true value of the measurand, the starting models become:$$I_{jk}=\mu_{jk}\left(1+S_{Ij}+R_{Ijk}\right)\;\;\;\;\;\;\;O_{jk}=\mu_{jk}\left(1+S_{Oj}+R_{Ojk}\right)$$
where $S_{*j}∼N\left(0,\sigma_{S*}^{2}\right)$ and $R_{*jk}∼N\left(0,\sigma_{R*}^{2}\right)$. In this case, the next step is defining the relative difference, given by:(2)$$\frac{O_{jk}-I_{jk}}{\mu_{jk}}=\frac{\mu_{jk}\left(1+S_{Oj}+R_{Ojk}\right)-\mu_{jk}\left(1+S_{Ij}+R_{Ijk}\right)}{\mu_{jk}}=S_{j}+R_{jk}.$$ This expression is again a random effect model, with variance components equal to $\sigma_{S}^{2}=\sigma_{SO}^{2}+\sigma_{SI}^{2}$ and $\sigma_{R}^{2}=\sigma_{RO}^{2}+\sigma_{RI}^{2}$. Since the measurand $\mu_{jk}$ is unknown, a reasonable solution is to substitute it with $O_{jk}$, because the operator measurement is typically more accurate and precise than the inspector’s measurements [7]. Ref. [12] shows that (i) assuming a truncated normal distribution for $O_{jk}$ that guarantees finite moments for the ratio $\frac{O_{jk}-I_{jk}}{O_{jk}}$, and (ii) considering that typically $\sqrt{\sigma_{SO}^{2}+\sigma_{RO}^{2}}\leq 0.02$ and $\sqrt{\sigma_{SI}^{2}+\sigma_{RI}^{2}}\leq 0.05$, we have that:-the distribution of the ratio $\frac{O_{jk}-I_{jk}}{O_{jk}}$ is extremely close to a normal distribution;-it provides an accurate approximation of the target variable.

Expressions (Equation 1) and (Equation 2) represent both a balanced one-way random effect models. The former assumes no relation between the magnitude of the measurand $\mu_{jk}$ and its corresponding error, whereas in the later the error is proportional to the measurand. Since nuclear measurements often have larger uncertainty at larger true values, a multiplicative rather than an additive model is employed [9]. It is also possible to transform a multiplicative model into an additive one through the well-known approximation log(1+x)≈x. However, this solution is not recommended due to the inaccuracies it could generate in many situations [2].

The parameters that need to be estimated in both (Equation 1) and (Equation 2) are the two variance components: the short-term systematic and the random one. In order to extend our focus also to unbalanced cases, in what follows, we will refer to the general formulation of the unbalanced one-way random effect model:(3)$$y_{jk}=\mu +\varsigma_{j}+\rho_{jk}\;\;\;\;k=1,\cdots ,n_{j}\;\;\;j=1,\cdots ,g$$
where $\varsigma_{j}∼N\left(0,\sigma_{\varsigma }^{2}\right)$ and $\rho_{jk}∼N\left(0,\sigma_{\rho }^{2}\right)$. Therefore, we have that:(4)$$\mathrm{Cov}\left(y_{jk},y_{lm}\right)=\left\{\begin{array}{cc} \sigma_{\tau }^{2}=\sigma_{\varsigma }^{2}+\sigma_{\rho }^{2} & \mathrm{if}\;j=l\;\mathrm{and}\;k=m\\ \sigma_{\varsigma }^{2} & \mathrm{if}\;j=l\;\mathrm{and}\;k\neq m\\ 0 & \mathrm{if}\;j\neq l \end{array}\right.$$ Classical estimators for $\sigma_{\varsigma }^{2}$ and $\sigma_{\rho }^{2}$ are given by [15]:(5)$$\begin{array}{c} {\hat{\sigma }}_{\rho }^{2}=\frac{\sum_{j=1}^{g}\sum_{k=1}^{n_{j}}\left(y_{jk}-{\bar{y}}_{j\cdot }\right)}{N-g}\\ {\hat{\sigma }}_{\varsigma }^{2}=\frac{\sum_{j=1}^{g}\left({\bar{y}}_{j\cdot }-{\bar{y}}_{\cdot \cdot }\right)}{g}-\bar{n}{\hat{\sigma }}_{\rho }^{2} \end{array}$$
where:$${\bar{y}}_{j\cdot }=\frac{\sum_{k=1}^{n_{j}}y_{jk}}{n_{j}}\;\;\;\;\;{\bar{y}}_{\cdot \cdot }=\frac{\sum_{j=1}^{g}{\bar{y}}_{j\cdot }}{g}\;\;\;\;\;N=\sum_{j=1}^{g}n_{j}\;\;\;\;\;\bar{n}=\frac{\sum_{j=1}^{g}n_{j}^{-1}}{g}$$
when $n_{j}=n$ for all j=1,⋯,g, expressions (Equation 5) simplify to the well-known ANOVA estimators for balanced design.

Finally, it could happen that ${\hat{\sigma }}_{\varsigma }^{2}<0$. In this case, the usual practice is to put ${\hat{\sigma }}_{\varsigma }^{2}=0$, that implies ${\hat{\sigma }}_{\tau }^{2}={\hat{\sigma }}_{\rho }^{2}$. In nuclear safeguards contexts, this case represents the situation where there are no significant differences between groups of measurements, and all the variability is due to the within groups component.

## 3. Robust Estimation of Unbalanced One-Way Random Effect Models

Classical estimation of model (Equation 3) through the expressions (Equation 5) yields results with desirable statistical properties whenever no anomalous observations are present in the data. When this assumption is not reliable, the use of robust statistical methods allows to avoid the bias introduced by the contaminated observations. In particular, two different kinds of outlier may affect one-way random effect models:(a)*Anomalous values of the group error*$\varsigma_{j}$, also called outlying blocks;(b)*Anomalous values of the individual measurement error*$\rho_{jk}$, also called outlying measurements within blocks.

This double nature of contamination is reflected also in a double definition of breakdown point, defined in general as the smallest proportion of outlying observations that can lead to estimated values which tend to 0 or *∞*. Following the definitions provided by [16], we will define the *block breakdown point* for contamination (a) and *measurement breakdown point* for contamination (b). Actually, [17] proposed the definition of a third outlier category, that will be not considered in detail, represented by an extremely small or large variation within one group.

In statistical terms, the price to pay when using robust estimators is a loss of efficiency, given that they are in general less efficient than their classical counterparts. However, an accurate choice of the robust method could sensibly mitigate this drawback, allowing to select the robust estimator that offers the most convenient payoff between breakdown point and efficiency. At this proposal, different choices are available for robustly estimating $\sigma_{\tau }$ and $\sigma_{\rho }$: from the M-estimators described in [18,19,20,21] to the S-estimator proposed by [22] and the GM estimator developed by [23]. Among all these options, we decided to consider the estimators proposed by [13] and based on the Q-estimator for one-sample scale introduced by [14]. The definition of the estimators requires the introduction of the following sets:$$\begin{array}{c} D_{between}\equiv \left\{\right|y_{jk}-y_{lm}\left|,1\leq j<l\leq g,1\leq k\leq n_{j},1\leq m\leq n_{l}\right\}\\ D_{within}\equiv \left\{\right|y_{jk}-y_{jm}\left|,1\leq j\leq g,1\leq k<m\leq n_{j}\right\}\\ D_{+}^{2}\equiv \left\{\right|y_{jk}-y_{js}+y_{lm}-y_{lh}\left|,1\leq j<l\leq g;1\leq k<s\leq n_{j};1\leq m<h\leq n_{l}\right\}\\ D_{-}^{2}\equiv \left\{\right|y_{jk}-y_{js}-y_{lm}+y_{lh}\left|,1\leq j<l\leq g;1\leq k<s\leq n_{j};1\leq m<h\leq n_{l}\right\} \end{array}$$
$D_{between}$ and $D_{within}$ are respectively the sets of the absolute intragroup and intergroup differences, whereas the union of $D_{+}^{2}$ and $D_{-}^{2}$ represents the set of the absolute second order differences of each group pair. The calculation of the estimators is based on the first or second quartile of these two numerical sets. The robust estimator for $\sigma_{\tau }$ is given by:(6)$${\tilde{\sigma }}_{\tau }=2.2191{\left\{d:d\in D_{between}\right\}}_{\left(0.25\right)}$$
whereas the two alternative proposals for estimating $\sigma_{\rho }$ are:(7)$$\begin{array}{c} {\tilde{\sigma }}_{\rho ,1}=1.0484{\left\{d:d\in D_{within}\right\}}_{\left(0.50\right)}\\ {\tilde{\sigma }}_{\rho ,2}=1.5692{\left\{d:d\in D_{+}^{2}\vee d\in D_{-}^{2}\right\}}_{\left(0.25\right)} \end{array}$$
where the expression $A_{\left(p\right)}$ represents the *p*-percentile of set A. The constant factors 2.2191, 1.0484 and 1.5692 guarantee Fisher consistency of the estimators when the data are normally distributed. Therefore, in applications with real data, a normality test for assessing the Gaussian assumption is recommended. Finally, as for the classical estimates, there is the possibility to obtain incoherent estimates, that is,
(8)$${\tilde{\sigma }}_{\rho ,*}>{\tilde{\sigma }}_{\tau }.$$ The solution in this case is again to put ${\tilde{\sigma }}_{\tau }={\tilde{\sigma }}_{\rho ,*}$.

The reasons that motivated the choice of these particular estimators are the following:

*Suitable breakdown point:* [16] proved that, in case of balanced model (i.e., $n_{j}=n$ for all j=1,⋯,g in expression (Equation 3)), the block breakdown points of ${\tilde{\sigma }}_{\tau }$, ${\tilde{\sigma }}_{\rho ,1}$ and ${\tilde{\sigma }}_{\rho ,2}$ are equal to ⌈g/2⌉/g, whereas their measurement breakdown points are asymptotically larger or equal to 0.25.

*High asymptotic efficiency:* defining the asymptotic efficiency as the ratio between the variances of asymptotic distributions of the robust estimators and those of the classical estimators, [16] proved that the asymptotic efficiency of ${\tilde{\sigma }}_{\tau }$ and ${\tilde{\sigma }}_{\rho ,2}$ in a balanced experiment is at least 82%, and can even exceed 90% depending on the values of *n* and $\sigma_{\varsigma }/\sigma_{\rho }$. The asymptotic efficiency of ${\tilde{\sigma }}_{\rho ,1}$, instead, starts at 37% for n=2 and increase up to 86% when *n* increases.*Closed expression:*${\tilde{\sigma }}_{\tau }$, ${\tilde{\sigma }}_{\rho ,1}$ and ${\tilde{\sigma }}_{\rho ,2}$ are the direct outcome of an explicit formula applied on a set of data. This means that they are not the result of iterative procedures and/or optimization algorithms that may be difficult to calculate and, more importantly, rather time consuming. The only complication of the proposed estimators that might lead to a sensible raise in the execution time of quartile calculation is the growth of the cardinality of $D_{+}^{2}$ and $D_{-}^{2}$ when *N* and *g* increase. However, for moderate values of *N* and *g* as the ones considered in this article, this does not seem to be an issue. Moreover, ${\tilde{\sigma }}_{\tau }$, ${\tilde{\sigma }}_{\rho ,1}$ and ${\tilde{\sigma }}_{\rho ,2}$ do not depend on the estimation of any location parameter.

## 4. Small Sample Comparison of Classical and Robust Estimates

In this section, we want to study and compare through a simulation exercise the behavior of the classical and robust variance estimators defined in (Equation 5)–(Equation 7). At this proposal, we will firstly simulate random effect models according to (Equation 3) without any kind of contamination. The objective of this first experiment is twofold. From one side, we want to compare the behavior of classical and robust estimates in an ideal empirical environment. From the other, it represents a benchmark for a fair assessment of the effects of outliers on both estimates in the second set of simulations that involves contaminated samples.

In order to have an exhaustive view on the main patterns that characterize the estimators in a finite-sample environment, the contaminated and uncontaminated simulations will consider different combinations of settings regarding:(i)the total number of observations $N=\sum_{j=1}^{g}n_{j}$;(ii)the number of groups *g*, that in nuclear safeguards context corresponds to the number of calibration or inspection periods;(iii)balanced and unbalanced samples. The values of $n_{j}$ that describe the unbalanced cases are provided in Table 1;(iv)the ratio between the systematic variance component and the total variance: $\psi^{2}=\sigma_{\varsigma }^{2}/\sigma_{\tau }^{2}$. Actually, looking at expression (Equation 4), the value of $\psi^{2}$ represents the correlation between observations in the same group. Fixing the value of the total variance to 1 (that is, $\sigma_{\tau }^{2}=\sigma_{\rho }^{2}+\sigma_{\varsigma }^{2}=1$), we will consider three different cases:-$\psi^{2}=0.25$, that implies $\sigma_{\varsigma }=0.500$ and $\sigma_{\rho }=0.866$;-$\psi^{2}=0.5$, that implies $\sigma_{\rho }=\sigma_{\varsigma }=0.707$;-$\psi^{2}=0.75$, that implies $\sigma_{\varsigma }=0.866$ and $\sigma_{\rho }=0.500$.

Similar to [23], in each simulated sample, we considered three different contamination schemes:(A)contamination of 10% of the random errors $\rho_{jk}$ in different groups replaced by $6\sigma_{\rho }$;(B)contamination of 10% of the short-term systematic errors $\varsigma_{j}$, replaced by $6\sigma_{\varsigma }$. For this scheme, we skip the case g=3, and when g=6, we contaminate one systematic error in each simulation, so slightly more than 10%;(C)10% of the random errors $\rho_{jk}$ in different groups replaced by $6\sigma_{\rho }$ and at least 10% of the short-term systematic errors $\varsigma_{j}$, whose random errors are not contaminated, replaced by $6\sigma_{\varsigma }$. Whenever possible, the two contamination schemes do not involve the same group. As before, we skip the case g=3, and when g=6 we contaminate one systematic error in each simulation, so slightly more than 10%;

The programming environment used for the simulation exercise is MATLAB (release R2021a). The codes for running and replicating our experiment are available upon request. They are described in the Appendix A, where we also provide all the implementation details.

Before presenting the results, it is important to notice that condition (Equation 8) may lead to two set of estimates for ${\tilde{\sigma }}_{\tau }$, one connected to ${\tilde{\sigma }}_{\rho ,1}$, and a second connected to ${\tilde{\sigma }}_{\rho ,2}$. However, the simulations outcomes did not highlight relevant differences in the average behavior of the two sets. Therefore, only the results obtained on the first will be presented.

**Table 1 entropy-24-01160-t001:** List of the $n_{j}$ values in unbalanced samples.

*N*	*g*	$n_{j}$
30	3	4,10,16
30	6	3,3,5,5,7,7
60	6	4,4,8,12,16,16
60	10	4,4,5,5,6,6,7,7,8,8
100	10	4,4,7,7,10,10,13,13,16,16
100	20	3,3,3,3,4,4,4,4,5,5,5,5,6,6,6,6,7,7,7,7
200	20	4,4,4,4,7,7,7,7,10,10,10,10,13,13,13,13,16,16,16,16
200	40	3,3,3,3,3,3,3,3,4,4,4,4,4,4,4,4,5,5,5,5,5,5,5,5,6,6,6,6,6,6,6,6,7,7,7,7,7,7,7,7

### 4.1. Simulation Results—No Contamination

Figure 2 presents the values of the Mean Absolute Bias (MAB) multiplied by 100 of the classical and robust variance components estimates obtained in the 100,000 simulations. As expected, in all cases, the absolute biases sensibly decrease on average when *N* and *g* increase. Similar patterns characterize the values of both classical and robust estimates, with the only difference that the classical estimates always offer better performances than the robust ones. There are only marginal differences between the MAB in balanced and unbalanced samples, with the former slightly outperforming the latter. The value of $\psi^{2}$ has, instead, a remarkable effect on the outcomes. Increasing the value of $\psi^{2}$ yields always to worst estimates of $\sigma_{\tau }$ and a more accurate estimation of $\sigma_{\rho }$. The pattern of MAB for $\sigma_{\rho }$ is also affected by the values of *N* and *g*: increasing *g* (for a fixed *N*) worsens the accuracy of classical and robust estimators, whereas the opposite happens if we increase *N* (for a fixed *g*). The estimates of $\sigma_{\tau }$, instead, seem to be only marginally influenced by the combination of *N* and *g*. Finally, it is worth pointing out that ${\tilde{\sigma }}_{\rho ,2}$ provides always more accurate estimates on average than ${\tilde{\sigma }}_{\rho ,1}$.

Figure 3 shows the estimated efficiencies of ${\tilde{\sigma }}_{\tau }$, ${\tilde{\sigma }}_{\rho ,1}$ and ${\tilde{\sigma }}_{\rho ,2}$. Following [24], in order to obtain a natural measure of accuracy of the estimator, the calculation of the efficiency is based on the standardized variance that, for a generic estimator $S_{M}$, can be calculated as:(9)$$\eta \left(S_{M}\right)=M\times \mathrm{Var}\left(S_{M}\right)/\mathrm{Mean}{\left(S_{M}\right)}^{2}$$
where Mean$\left(S_{M}\right)$ and Var$\left(S_{M}\right)$ refer to the values obtained from the simulations. The ratio $\eta \left({\hat{\sigma }}_{*}\right)/\eta \left({\tilde{\sigma }}_{*}\right)$ between the standardized variances of the classical and the robust estimator will then provide an estimate of the efficiency. Coherently to what proved in [16], the panels of Figure 3 confirm that ${\tilde{\sigma }}_{\rho ,1}$ is less efficient than ${\tilde{\sigma }}_{\rho ,2}$, and that the efficiencies of both do not depend in general on the ratio $\psi^{2}$, or on the values of *N* and *g*. Even in small samples, the estimated efficiency of ${\tilde{\sigma }}_{\rho ,2}$ is about 95% in balanced samples and 90% in unbalanced samples, whereas that one of ${\tilde{\sigma }}_{\rho ,1}$ oscillates around 65% in both cases. The estimated efficiency of ${\tilde{\sigma }}_{\tau }$, instead, varies from 30% (when *N* and *g* are small) to 85% (for larger values of *N* and *g*). The value of $\psi^{2}$ has a marginal effect, whereas remarkable differences characterize the value of the estimated efficiency of ${\tilde{\sigma }}_{\tau }$ in balanced and unbalanced samples.

Concluding, simulations results on noncontaminated data provided a comprehensive description of the finite-sample properties of classical and robust estimators of variance components. Even though classical estimates resulted more accurate on average, the gap with respect to the respective robust counterparts is limited. Also in terms of efficiency, the choice of the robust alternative does not imply a remarkable loss, especially for $\sigma_{\rho }$.

### 4.2. Simulation Results—Contaminated Samples

Figure 4 shows the effects of contamination (A) in terms of MAB on the classical and robust estimates, and give the possibility to compare these results with the benchmark case of no contamination. As expected, both figures highlight a remarkable increase of the estimated bias of all estimators. However, robust estimators have the capacity to limit the distortion yielded by outlying observations, showing more accuracy than classical estimators. This is particularly evident for the estimates of $\sigma_{\rho }$ independently of the simulation settings. As expected, the bigger efficiency of ${\tilde{\sigma }}_{\rho ,2}$ with respect to ${\tilde{\sigma }}_{\rho ,1}$ highlighted in Figure 3 is counterbalanced by a smaller accuracy in contaminated contexts. Concerning the estimates of $\sigma_{\tau }$, the robust estimator clearly show more accurate outcomes when $\psi^{2}<0.75$, whereas when $\psi^{2}=0.75$, ${\tilde{\sigma }}_{\tau }$ is slightly outperformed by ${\hat{\sigma }}_{\tau }$ only in small samples (i.e., N=30). However, in empirical application on nuclear measurement data, usually $\psi^{2}\leq 0.5$, given that $\sigma_{\varsigma }$ tends to be smaller than $\sigma_{\rho }$ [10].

Results concerning contamination (B) are displayed in Figure 5. Since the contamination involves only the systematic component of model (Equation 2), both classical and robust estimates of $\sigma_{\rho }$ are not affected by anomalous data and confirm the same results previously observed in Figure 2b. Their MAB values are almost the same, and this explains why in panel Figure 5b the grey lines (representing the results on contaminated samples) are indistinguishable from the black lines (representing the results on samples without contamination). ${\hat{\sigma }}_{\tau }$ and ${\tilde{\sigma }}_{\tau }$, instead, worsen their performances due to contamination. However, once again, the robust version of the estimator limits the negative effects of the outliers and provides more accurate estimates. This is particularly true when the samples are balanced, and when $\psi^{2}\geq 0.50$ in unbalanced designs.

Finally, Figure 6 presents the simulation results obtained for contamination (C). The best results in terms of MAB offered by the robust estimators are quite evident. The two robust estimators of $\sigma_{\rho }$ always provide remarkably smaller values of MAB than the classical one, with again ${\tilde{\sigma }}_{\rho ,1}$ outperforming ${\tilde{\sigma }}_{\rho ,2}$. The same can be said also for ${\tilde{\sigma }}_{\tau }$, with the only exception of small unbalanced samples, where the two estimators yield approximately the same outcome.

Figure 7 shows in detail the simulated observations for a balanced sample of N=60 observations in g=6 groups under contamination (A). These data may mimic the situation where an inspector has to assess the measurements obtained in the sixth inspection period. The assessment is usually based on the outcomes obtained in the previous five inspections, and on the 3σ rule for the declaration of the outliers. This common practice implicitly assumes that previous measurements are free from anomalies. The plot clearly shows that, if this assumption does not hold, the outlier in the period of interest could remain undetected. Robust estimators, instead, are calculated using also the values of $\frac{O-I}{O}$ that we aim to assess and, being only partially affected by past anomalous values, allow a correct identification of the outlier in the last inspection.

## 5. Empirical Application

Figure 8 presents the $\left(O_{jk}-I_{jk}\right)/O_{jk}$ values effectively registered in 6 inspection periods where impure Plutonium items were measured in attended mode. The 52 values form an unbalanced sample, since the number of measurements in each of the 6 inspection periods is 12, 8, 10, 10, 7 and 5. There are 4 values (highlighted in red) that are clearly far from the general pattern. Posterior cross check between the operator and inspection measurements led to a confirmation that those 4 values were inconsistent with the declarations and that the corresponding measurements were actually outliers.

Classical estimates obtained for $\sigma_{\tau }$, $\sigma_{\rho }$ and $\sigma_{\varsigma }$ are given respectively by 0.1315, 0.1133 and 0.0667. The robust method, instead, leads to ${\tilde{\sigma }}_{\rho ,1}=0.0894$, ${\tilde{\sigma }}_{\rho ,2}=0.0958$ and ${\tilde{\sigma }}_{\tau }=0.0869$. Therefore, in both cases, robust estimates satisfy condition Figure Equation 8, and then we need to put ${\tilde{\sigma }}_{\tau }={\tilde{\sigma }}_{\rho ,*}$, that in turn implies ${\tilde{\sigma }}_{\varsigma }=0$. Therefore, the presence of the 4 anomalous data had a twofold effect on the outcomes:classical estimates for $\sigma_{\tau }$ are always larger than robust ones;differently from classical one, robust estimation does not detect any relevant short-term systematic error component.

Coherently to what we observed in the simulations, the presence of outliers in the sample yields an overestimation in the variance components that the robust methods is capable to limit. The 3σ thresholds represented in the figure provide a further confirmation. The classical ones, based on the sample mean and on ${\hat{\sigma }}_{\tau }$, include almost all the outliers. This is due to the effect that the 4 anomalies has on the calculation of both the sample mean and the variance components. The robust ones, based on the median and on ${\tilde{\sigma }}_{\tau }$, are tighter and more centered with the good measurements, given that the median is a robust estimator of the central tendency, and its value is not affected by the anomalous data. As a result, the four outliers are all outside the robust 3σ thresholds.

## 6. Conclusions

The availability of an efficient and unbiased estimation of the uncertainty components in nuclear safety measurements is a fundamental element for a reliable detection of possible data falsification by operators that could mask diversion. Top-down methods for UQ rely fundamentally on modeling measurement data as a one-way random effect model with within and between groups variance components, which are usually estimated through the classical ANOVA expressions. However, the reliability of these estimates could be seriously affected by the presence of outliers in the data.

The aim of this article was to introduce the use of robust approaches in the UQ of nuclear safety measurements, and to analyze the general benefits in terms of estimates accuracy and bias reduction. For this purpose, we proposed a robust estimator of the variance components that combines several desirable statistical properties, i.e., suitable breakdown point, high efficiency, closed expression. Simulation results and the empirical application proved that the presence of even on single outlier may significantly affect the reliability of classical estimates. The robust estimators, instead, have the capacity to handle anomalous data, and to remarkably limit their negative effects on the final estimates.

Robust statistics offers many other alternative methods for estimating within and between group variance. The promising outcomes obtained through the simulation exercise and the empirical application may be considered as a starting point for a broader investigation aimed to compare different robust estimators for one-way random effect models in simulated and empirical contexts, and to identify the more suitable one. Such analysis will undoubtedly represent a solid contribution to the still ongoing development of reliable UQ methods for error variance estimation.

## Figures and Tables

**Figure 1 entropy-24-01160-f001:**
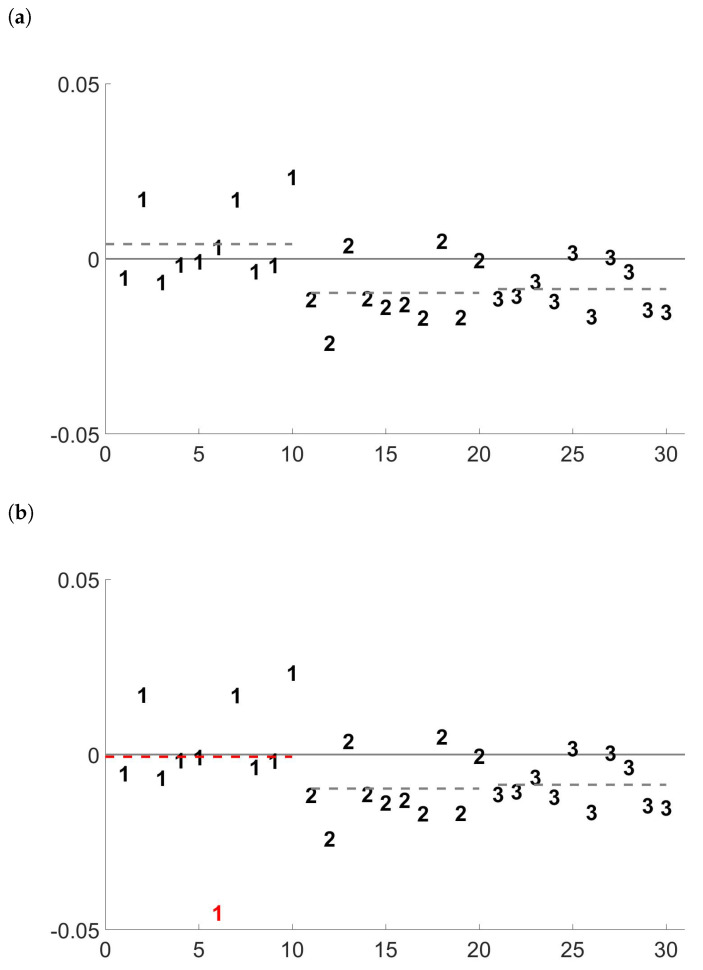
Effect of the introduction of one single outlier (highlighted in red) on the simulated case study of [12] (dashed lines represent the average of the 10 observations in each group). (**a**) Clean data; (**b**) One outlier.

**Figure 2 entropy-24-01160-f002:**
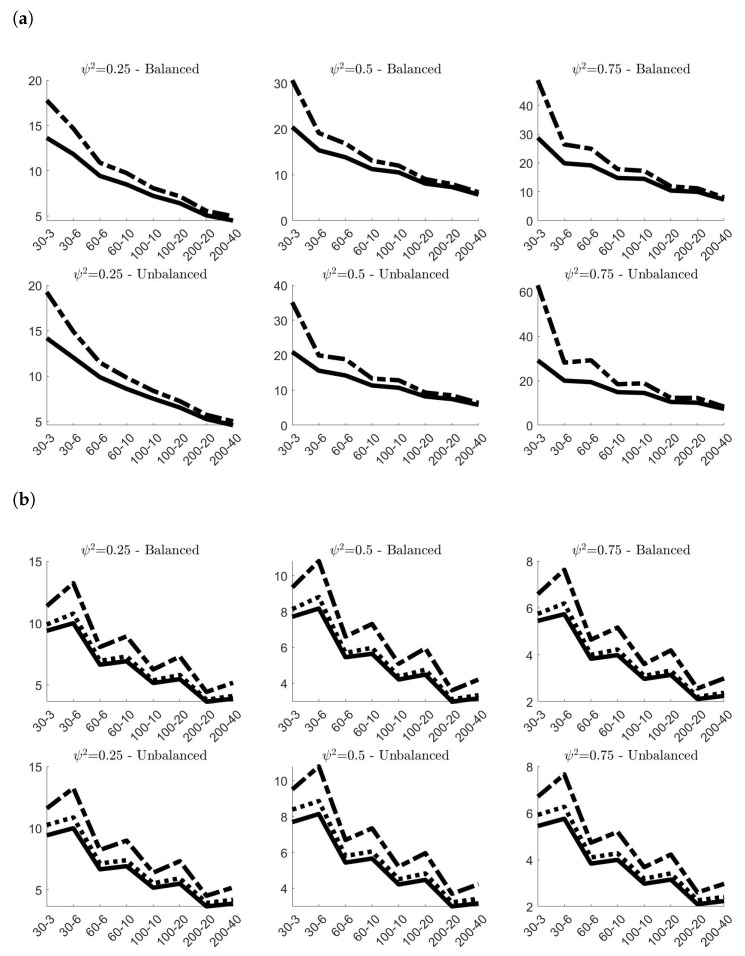
100×MAB of the variance components estimates obtained in the 100,000 simulations **without contamination**^*^. (**a**) 100×MAB of ${\hat{\sigma }}_{\tau }$ (classical estimator, solid line) and ${\tilde{\sigma }}_{\tau }$ (robust estimator, dashed line); (**b**) 100×MAB of ${\hat{\sigma }}_{\rho }$ (classical estimator, solid line), ${\tilde{\sigma }}_{\rho ,1}$ (robust estimator, dashed line) and ${\tilde{\sigma }}_{\rho ,2}$ (robust estimator, dotted line). ^*^ In all simulations $\sigma_{\tau }=1$. The labels on the *x*-axis are in the form *N*–*g*.

**Figure 3 entropy-24-01160-f003:**
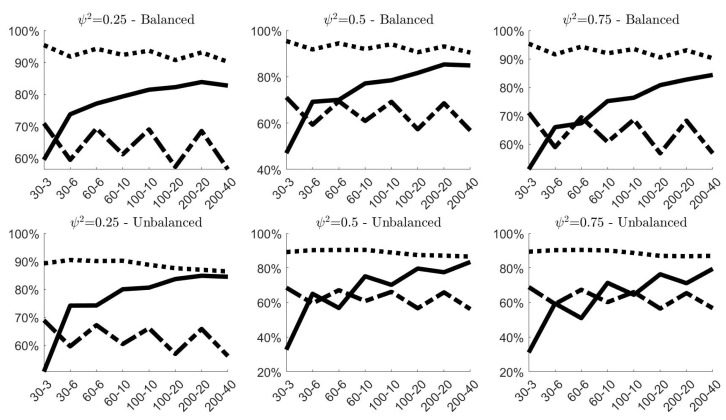
Estimated efficiency of the robust estimators ${\tilde{\sigma }}_{\tau }$ (solid line), ${\tilde{\sigma }}_{\rho ,1}$ (dashed line) and ${\tilde{\sigma }}_{\rho ,2}$ (dotted line) calculated on the 100,000 simulations **without contamination**^*^. ^*^ In all simulations $\sigma_{\tau }=1$. The labels on the *x*-axis are in the form *N*–*g*.

**Figure 4 entropy-24-01160-f004:**
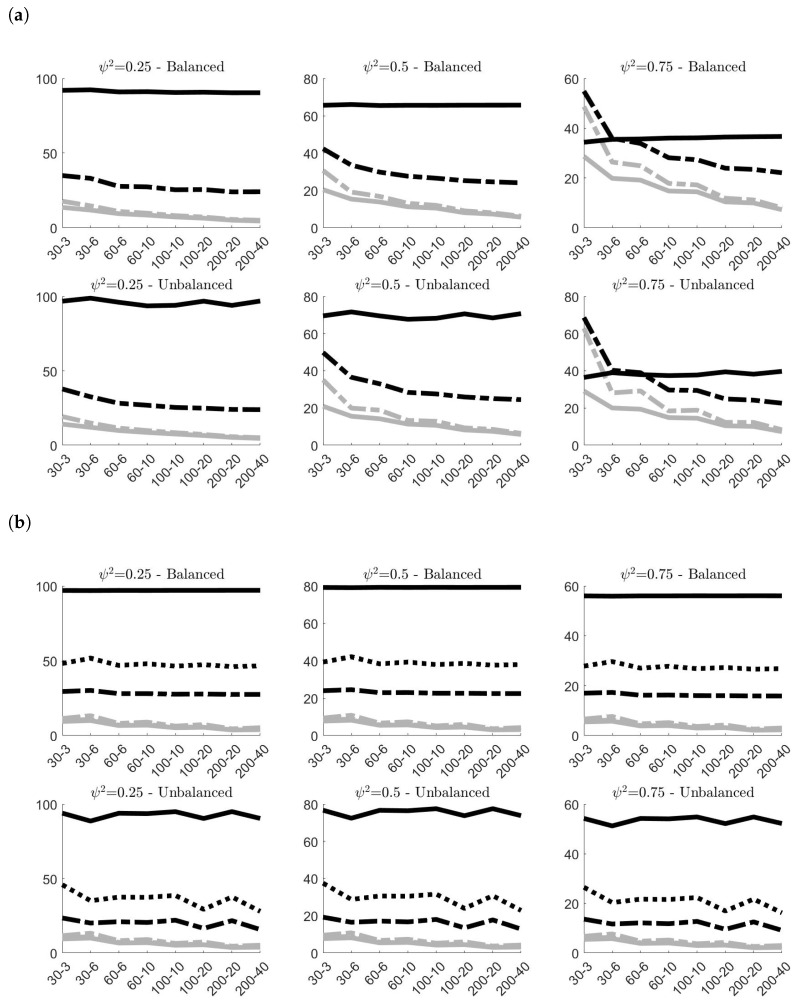
100× MAB of the variance components estimates obtained in the 100,000 simulations without contamination (gray lines) and under **contamination (A)** (black lines)^*^. (**a**) 100×MAB of ${\hat{\sigma }}_{\tau }$ (classical estimator, solid line) and ${\tilde{\sigma }}_{\tau }$ (robust estimator, dashed line); (**b**) 100×MAB of ${\hat{\sigma }}_{\rho }$ (classical estimator, solid line), ${\tilde{\sigma }}_{\rho ,1}$ (robust estimator, dashed line) and ${\tilde{\sigma }}_{\rho ,2}$ (robust estimator, dotted line). ^*^ In all simulations $\sigma_{\tau }=1$. The labels on the *x*-axis are in the form *N*–*g*.

**Figure 5 entropy-24-01160-f005:**
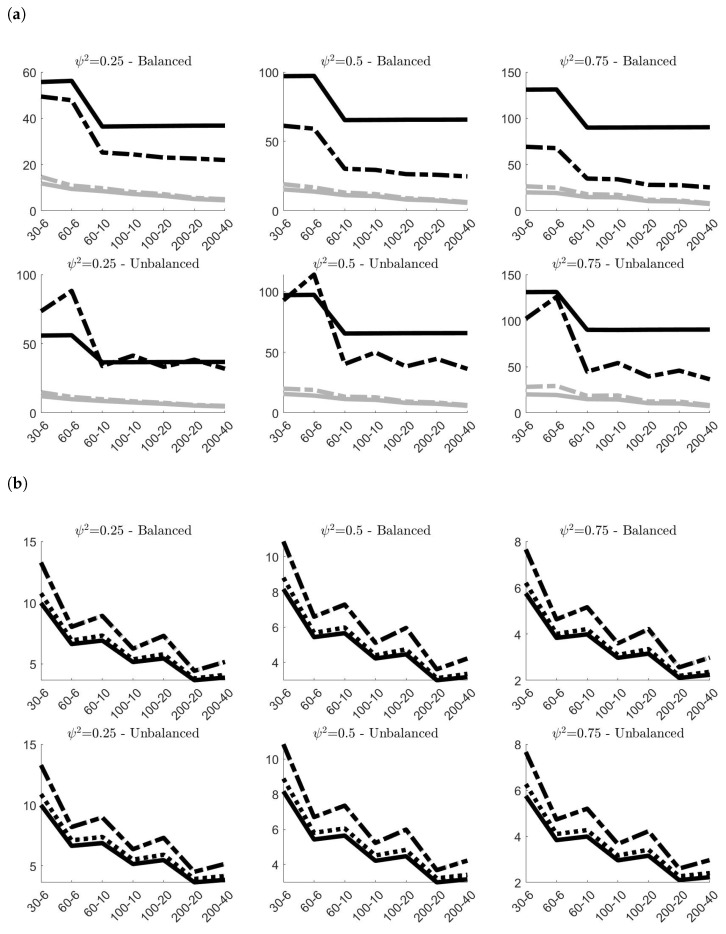
100×MAB of the variance components estimates obtained in the 100,000 simulations without contamination (gray lines) and under **contamination (B)** (black lines)^*^. In panel Figure 5b, black and grey lines overlap. (**a**) 100×MAB of ${\hat{\sigma }}_{\tau }$ (classical estimator, solid line) and ${\tilde{\sigma }}_{\tau }$ (robust estimator, dashed line); (**b**) 100×MAB of ${\hat{\sigma }}_{\rho }$ (classical estimator, solid line), ${\tilde{\sigma }}_{\rho ,1}$ (robust estimator, dashed line) and ${\tilde{\sigma }}_{\rho ,2}$ (robust estimator, dotted line). ^*^ In all simulations $\sigma_{\tau }=1$. The labels on the *x*-axis are in the form *N*–*g*.

**Figure 6 entropy-24-01160-f006:**
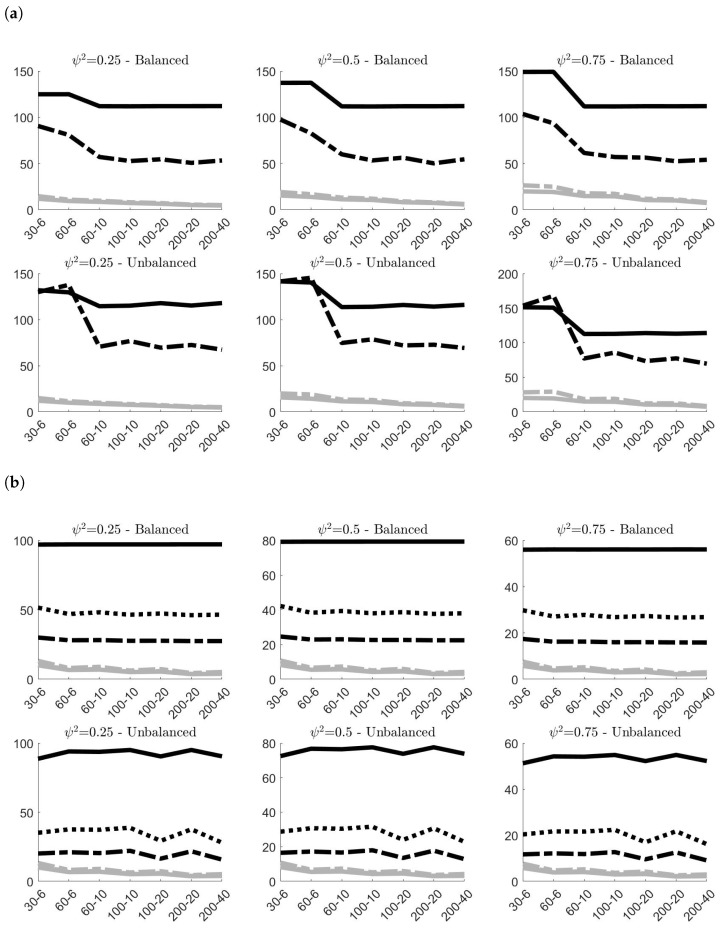
100×MAB of the variance components estimates obtained in the 100,000 simulations without contamination (gray lines) and under **contamination (C)** (black lines)^*^. (**a**) 100×MAB of ${\hat{\sigma }}_{\tau }$ (classical estimator, solid line) and ${\tilde{\sigma }}_{\tau }$ (robust estimator, dashed line); (**b**) 100×MAB of ${\hat{\sigma }}_{\rho }$ (classical estimator, solid line), ${\tilde{\sigma }}_{\rho ,1}$ (robust estimator, dashed line) and ${\tilde{\sigma }}_{\rho ,2}$ (robust estimator, dotted line). ^*^ In all simulations $\sigma_{\tau }=1$. The labels on the *x*-axis are in the form *N*–*g*.

**Figure 7 entropy-24-01160-f007:**
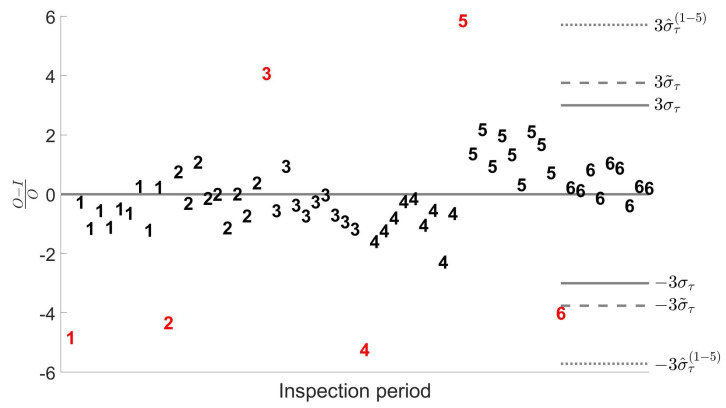
Simulated observations of a balanced sample with N=60 and g=6 under **contamination (A)**. **Solid line**: 3σ threshold calculated with the real value of the mean and of the total standard error $\sigma_{\tau }$. **Dotted line**: 3σ threshold based on the real value of the mean and on the classical estimation ${\hat{\sigma }}_{\tau }$ calculated with the data of the first five inspections. **Dashed line**: 3σ threshold based on the real value of the mean and on the robust estimation ${\tilde{\sigma }}_{\tau }$.

**Figure 8 entropy-24-01160-f008:**
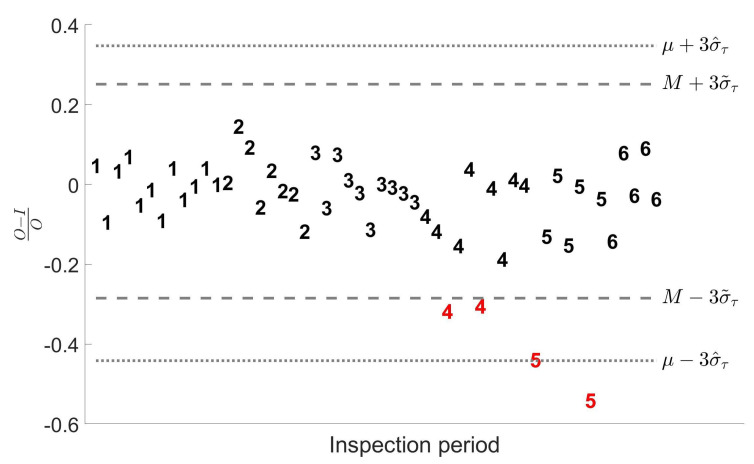
Classical and Robust estimation of $\sigma_{\tau }$ on a set of empirical data concerning impure Plutonium items measured in attended mode. **Dotted line**: 3σ threshold based on the sample mean and on the classical estimation ${\hat{\sigma }}_{\tau }$. **Dashed line**: 3σ threshold based on the sample median and on the robust estimation ${\tilde{\sigma }}_{\tau }={\tilde{\sigma }}_{\rho ,1}$.

## Data Availability

Not applicable.

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
