# Peer review of "Introducing Robust Statistics in the Uncertainty Quantification of Nuclear Safeguards Measurements"

_entropy, 2022, doi:10.3390/e24081160_

Round 1
Reviewer 1 Report
1. The language/grammar used is at times awkward, such as line 66
2. Authors should show their real data. It is unclear if authors deal with time-series data. In Figures 1a and 1b some time series data seems to be presented. Methods from times series literature can be employed to analyze such data and comparisons with the proposed methods can be explored.
3. It is unclear how authors picked the scalar values in expression (7) from lines 113-114. Generally, a data driven approach for parameters is typically preferred.
4. Software discussion and materials, implementation details and code needs to be presented.
Author Response
REPLIES TO REVIEWER 1
- The language/grammar used is at times awkward, such as line 66
Thanks a lot for your comment. The sentence at line 66 has been rephrased, and language/grammar of the whole article has been revised.
- Authors should show their real data. It is unclear if authors deal with time-series data. In Figures 1a and 1b some time series data seems to be presented. Methods from times series literature can be employed to analyze such data and comparisons with the proposed methods can be explored.
We agree with your comment. We changed the description of the figures. Now the characteristics of the data should be clearer. We also added the motivation that prevent from employing methods coming from time series literature in nuclear safeguards contexts. Inspections data cannot be considered as time series since the time lag between two single inspections and even between two groups of inspection is not fixed and constant. Finally, concerning the possibility to use and show real data, unfortunately nuclear measurements are quite sensible, and their values cannot be disseminated.
- It is unclear how authors picked the scalar values in expression (7) from lines 113-114. Generally, a data driven approach for parameters is typically preferred.
Thanks for raising this issue. In the revised version of the article, we added the motivations behind the choice of the constants in expressions (6) and (7). Concerning the use of data driven procedures, it could be surely object of future developments of the robust testing procedure adopted in this exercise.
- Software discussion and materials, implementation details and code needs to be presented.
Thanks for your suggestion. We added an appendix in the revised version of the manuscript where we describe the codes and the implementation details for replicating our results.

Reviewer 2 Report
See the attached review report.

Author Response
REPLIES TO REVIEWER 2
- Thanks for raising this issue. In the revised version of the article, we improved the explanation of the two models and provide details concerning the replacement of the unknown value with .
2, 3 and 4. We really thank the reviewer for these suggestions. The description of expressions (6) and (7) has been changed in order to consider them. We believe these changes remarkably improved the readability and the clearness of the manuscript.
- Thanks for this suggestion. We added a brief description of the practical implication of in the nuclear safeguards context.
- Thanks for the correction. It was clearly a mistake to state that the growth was “exponential”. This adjective has been removed from the revised version of the article.
- Thanks for raising this issue. Actually, the grey lines were present, but it was impossible to distinguish them from the black ones. This is because contamination (B) involves only the systematic component of model. Therefore, both classical and robust estimates of are not affected by anomalous data, and their MAB values are very similar. We detailed this issue in the revised version of the paper.
- Thanks for this further correction.
Round 2
Reviewer 2 Report
The author has addressed my comments and I recommend to accept the paper.
This manuscript is a resubmission of an earlier submission. The following is a list of the peer review reports and author responses from that submission.